# Environmental Analysis, Monitoring, and Process Control Strategy for Reduction of Greenhouse Gaseous Emissions in Thermochemical Reactions

Mohamed Aboughaly [1,*] and I. M. Rizwanul Fattah [2,*]

[1] Chemical Engineering Department, University of Saskatchewan, Management Area 3B48, 57 Campus Dr, Saskatoon, SK S7N 5A9, Canada
[2] Centre for Technology in Water and Wastewater (CTWW), School of Civil and Environmental Engineering, Faculty of Engineering and IT, University of Technology Sydney, Ultimo, NSW 2007, Australia
* Correspondence: mohamed.aboughaly@ontariotechu.net (M.A.); islammdrizwanul.fattah@uts.edu.au (I.M.R.F.)

**Abstract:** This review paper illustrates the recommended monitoring technologies for the detection of various greenhouse gaseous emissions for solid waste thermochemical reactions, including incineration, pyrolysis, and gasification. The illustrated gas analyzers are based on the absorption principle, which continuously measures the physicochemical properties of gaseous mixtures, including oxygen, carbon dioxide, carbon monoxide, hydrogen, and methane, during thermochemical reactions. This paper illustrates the recommended gas analyzers and process control tools for different thermochemical reactions and aims to recommend equipment to increase the sensitivity, linearity, and dynamics of various thermochemical reactions. The equipment achieves new levels of on-location, real-time analytical capability using FTIR analysis. The environmental assessment study includes inventory analysis, impact analysis, and sensitivity analysis to compare the mentioned solid waste chemical recycling methods in terms of greenhouse gaseous emissions, thermal efficiency, electrical efficiency, and sensitivity analysis. The environmental impact assessment compares each technology in terms of greenhouse gaseous emissions, including $CO_2$, $NO_x$, $NH_3$, $N_2O$, CO, $CH_4$, heat, and electricity generation. The conducted environmental assessment compares the mentioned technologies through 15 different emission-related impact categories, including climate change impact, ecosystem quality, and resource depletion. The continuously monitored process streams assure the online monitoring of gaseous products of thermochemical processes that enhance the quality of the end products and reduce undesired products, such as tar and char. This state-of-the-art monitoring and process control framework provides recommended analytical equipment and monitoring tools for different thermochemical reactions to optimize process parameters and reduce greenhouse gaseous emissions and undesired products.

**Keywords:** carbon monoxide; carbon dioxide; methane; syngas; gas analyzer; biomass; combustion; thermogravimetric analyzer; pyrolysis

## 1. Introduction

Monitoring equipment is used in research and development when the environmental footprint and emissions calculations make a significant impact on the process plant design, plant sizing, cost structure, and future marketing of the process technology [1]. Life cycle assessment strategies evaluate the environmental impact of greenhouse gaseous (GHG) emissions and ensure that the proposed process design complies with the environmental emission standards and regulations set by environmental agencies [2]. Life cycle assessment (LCA) also provides designers, environmental agencies, and engineers with options that are used in decision-making in different parts of the project, including the preliminary design, construction, or execution of chemical plants. LCA strategies are used in businesses

to optimize spending and comply with environmental regulations, as well as to compare alternative technologies in terms of spending and the carbon footprint of different process routes [3–5].

Thermochemical processes convert solid and plastic waste deposits, recover thermal energy, and generate electricity, as well as reduce environmental and health impacts [6,7]. This study assesses the different thermochemical processes over 15 different environmental indicators and techno-economic key performance indicators (KPI). This paper also provides a methodology for assessing the environmental impact of various thermochemical reactors for different chemical reactions and the selective comparison of environmental burdens. The environmental assessment considers greenhouse gaseous emissions as a comparative parameter for different thermochemical processes [8]. This paper also illustrates the methodology for assessing the environmental impact of greenhouse gaseous emissions for different scenarios of the previously mentioned thermochemical processes [9–11]. The assessed environmental impact focuses on the emissions of greenhouse gaseous products from various thermochemical reactions as a comparative parameter between the mentioned chemical reactions.

This paper provides a systematic framework to quantitively analyze the expected greenhouse gaseous emissions from various thermochemical reactions and the optimal temperature profiles to reduce them. Biomass-based production produces lower greenhouse gaseous emissions compared to solid and plastic waste thermochemical reactions [12–14].

The uncertainty of greenhouse gaseous emissions (GHG) is assessed using Monte Carlo simulation (MCS) and parameter estimation techniques. Treating a coefficient (i.e., the GHG emission factor) as a variable yields a higher uncertainty of greenhouse gaseous emissions compared to considering it as a coefficient constant. The parametric estimation techniques improve thermochemical processes and eliminate undesired products. The non-parametric bootstrap method improves reaction kinetics and eliminates greenhouse gaseous emissions. In case the estimated probability density function (PDF) is inaccurate, the non-parametric bootstrap method is used to assess the undesired products, including greenhouse gaseous emissions [13,15–18].

Using conservative assumptions (25% conversion and high energy separation), process optimization can reduce greenhouse gaseous emissions, causing emission reduction by as much as 94% [10,19–24]. The constraints on the fraction of chemicals, such as $CO_2$, methane, $NO_x$, and other gaseous emissions, from thermochemical reactions are the main contributors to global climate change. Carbon dioxide is considered the greenhouse gas with the highest contribution to global climate change [21,24,25].

To maintain an effective thermochemical reaction, it is vital to control and monitor all the dynamics in real time to evaluate the performance of the thermochemical conversion, as well as to reduce tar and char during the process [23,24]. The monitoring and process controlling equipment is used to monitor the dynamics of the chemical process and provide instantaneous feedback to increase the hydrocarbon yield and eliminate tar emissions [26–29].

Landfilling is responsible for the release of high quantities of methane gas generated from large quantities of biomaterials in the landfill. Countries such as Sweden impose a landfill tax to reduce landfilling activities, causing the elimination of such activities in 2005. Landfill gas consists of carbon monoxide and methane, which manifest a significant increase in acidification potential (i.e., the release of $SO_2$) and global warming potentials (i.e., the release of $CO_2$) [30,31]. Small countries with limited land space and dense populations have relied on chemical recycling activities as the preferred waste management strategy [1–20]. Thermochemical processes, such as pyrolysis and gasification, exhibit a lower carbon footprint, accessed by global warming potential and human toxicity measurement potential, as shown below in Figure 1 [32–35]:

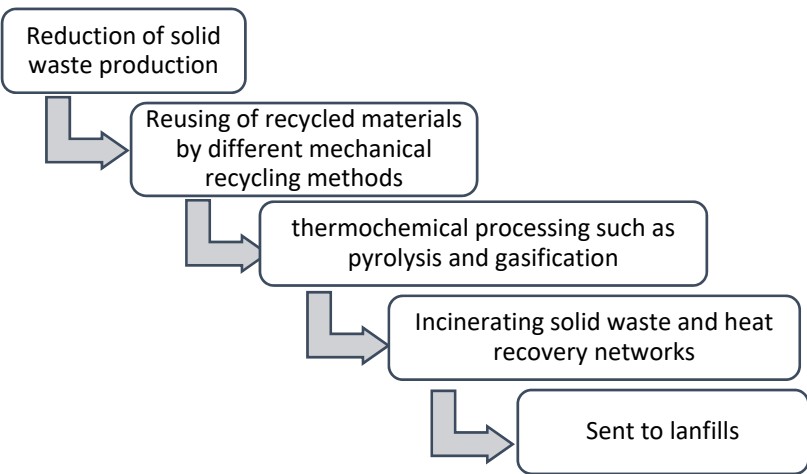

**Figure 1.** Hierarchy for global waste management techniques [36].

As seen in Figure 1, different strategies that eliminate or reduce municipal solid waste (MSW) are given priority. This includes designing durable manufacturing materials that help prevent the generation of solid waste. These strategies also encompass different waste control strategies in industrial activities and limiting the use of plastics, paper, and composts, as well as implementing internal recycling activities at manufacturing sites [37–39]. The second strategy includes reusing materials through recycling activities, including mechanical recycling and the extraction of the raw material for reuse using various mechanical processes, including extrusion, crushing, and pressing into raw materials to be used in new products. The third strategy includes thermochemical reactions, such as pyrolysis and gasification, which recycle solid and plastic waste into hydrocarbon products for energy generation [9,11,19,40]. Municipal solid waste landfilling is not recommended as a first waste management strategy due to its inherent high methane release, possible pollution of the soil, and effects on habitat life [41,42]. Landfilling is discouraged due to high land space requirements, soil erosion, and high operational and maintenance costs.

The development of sustainable and clean energy technologies reduces reliance on fossil fuels and decreases greenhouse gaseous emissions, which also aids in solid waste elimination and the generation of alternative fuels, including syngas and pyrolytic oil, and avoids incineration and landfilling activities [43,44]. The most practiced waste management strategy in North America is landfilling, and more than 50% of unprocessed solid waste is sent to landfills [6,37]. Pyrolysis and gasification provide higher energy production and help in the reduction of incineration and landfilling practices, as well as reduce the gas release of toxins, such as dioxins, $NO_x$, and CO emissions. They also provide higher electricity generation and support the production of gaseous and liquid hydrocarbon fuels [45]. Incineration provides the highest solid waste volume reduction, 90%, as well as high thermal energy recovery from waste. Incineration produces heat energy and electrical energy in the range of 600 to 700 KWh of electricity per ton of solid waste. Solid waste tar or ash can be converted to syngas, which includes $CO_2$ in the range of 12% to 15% [37,46]. Incinerators utilize the heat generated from the combustion of solid waste using heat recovery networks for steam generation [37,38]. Incineration is not recommended due to the high carbon dioxide generation of around 1.1 tons of carbon dioxide per ton of incinerated solid waste, as well as more than 390 g of $SO_2$, 1790 g of CO, and more than 850 g of $NO_2$, but which could generate more than 0.95 MWh of power under optimal incineration and the full combustion of solid waste [38,39].

Fast pyrolysis is a comparable technology to solid waste gasification due to its very high temperatures and fast residence time. Table 1 shows the product composition obtained from fast pyrolysis and gasification. It is recommended to use fast pyrolysis for the production of oil, syngas, or electricity.

**Table 1.** Products from pyrolysis and gasification [1–25].

| Chemical Process | Hydrocarbon Oils | Char (Solid) | Product Gas | Conversion Efficiency (%) | Electrical Efficiency (%) |
|---|---|---|---|---|---|
| Fast pyrolysis | 60–70% | 10–15% | 10–25% | 25–45 | 15 |
| Gasification | ≈20% | ≈20% | ≈85% | 26–40 | 34 |

Municipal solid waste thermal treatment methods depend on operational factors such as thermal efficiency, economic benefits, and environmental emissions. In the process of gasification, most syngas produced contains highly toxic and corrosive species, thus requiring syngas cleaning before combustion [37,38]. The key performance indicators are used to evaluate the expected final products and gaseous effluents emitted into the atmosphere. Syngas per ton of solid waste varies from 0.8 to 1.1 nm$^3$/kg, depending on the chemical composition. As seen below in Tables 2 and 3, to achieve higher absorption efficiency, multiple scrubbing beds in series might be required for $SO_2$, $NO_2$, and HCl to increase pollutant removal efficiency [47]. In Table 3, the environmental emissions from pyrolysis and gasification systems are illustrated. It can be noted that pyrolysis shows the lowest $NO_2$ and CO emissions in comparison with gasification, as shown in Table 4. Moreover, the pretreatment of MSW to RDF in gasification reactions reduces $SO_2$, CO, HF, and dust emissions, as shown.

**Table 2.** Operational methods for removal of syngas pollutants.

| Pollutant Gas | Operational Methods | Absorption Efficiency (%) |
|---|---|---|
| $SO_2$ | Wet scrubbing or dry cyclone | 55–95 |
| HCl | Wet or semi-dry scrubbing | 70–90 |
| $NO_2$ | Catalytic reduction | 15–63 |
| Heavy metals | Dry scrubbing and electrostatic precipitators | 75–95 |
| Ash | Carbon filters and optimum operating conditions | >95 |
| Dioxins and furans | Activated carbon beds and filters | 50–90 |

**Table 3.** Required absorption efficiency for syngas cleaning systems.

| Pollutant Gas | Maximum Allowable at the Exhaust (mg/Nm$^3$) | Required Removal Efficiency (%) |
|---|---|---|
| Ash | 10 | 99.9 |
| HCl | 10 | >99 |
| $SO_2$ | 5 | 99.5 |
| $NO_x$ | 70 | 86 |
| HF | 1 | 96 |
| Hg | 0.01 | 99 |
| Dioxins | 0.1 | 98 |

**Table 4.** Expected greenhouse gaseous emissions from pyrolysis and gasification [1–20].

| Gaseous Emission | Pyrolysis (g/Tonne) | Gasification of Non-RDF (g/Tonne) | Gasification of RDF (g/Tonne) |
|---|---|---|---|
| Nitrogen oxide | 196 | 781 | 453 |
| Sulphur dioxide | 28 | 19.5 | 10.5 |
| Carbon monoxide | 28.1 | 195 | 116 |
| Hydrogen chloride | 5.62 | 3.91 | 17.2 |
| Hydrogen fluoride | 0.562 | 3.91 | 0.116 |
| Dust | 8.43 | 39.1 | 6.97 |

The thermal efficiencies of gasifiers and pyrolyzers for solid waste reactors are calculated using the following equations [1–15]

$$\text{Gasifier energy efficiency (\%)} = \frac{(\text{Esyngas} + \text{Etar})}{\text{LHVmsw}} \times 100\% \qquad (1)$$

$$\text{Pyrolysis energy efficiency (\%)} = \frac{(\text{Eoil} + \text{Etar})}{\text{LHVmsw}} \times 100\% \qquad (2)$$

*Incineration and Landfilling Industrial Stages*

The most common practices of chemical recycling include incineration and landfilling in most countries that have a limited landscape or difficult free land accessibility, such as Japan, along with several European countries, including Germany and Finland, where incineration is used for district heating in which more than half of all solid waste is sent to incinerators [6,48]. Landfilling yields high methane release and long environmental impact compared to other thermochemical processes, as well as the highest release of methane, which is a greenhouse gas contributor released by the thermal decomposition of biomaterials by an anaerobic digestion process. Landfilling causes ground pollution from leachates, while incineration causes the release of $CO_2$, in which 75% of waste is converted into ashes, metals, and unburned combustibles [49,50]. Ash from incineration can be avoided by maintaining constant temperature profiles, MSW pretreatments, and stoichiometric oxygen flow rates. Below are listed the expected final products from incinerated municipal solid waste, as shown in Table 5 [42,45,51]:

**Table 5.** Expected final products from incinerated municipal solid waste deposits [1–20].

| MSW (%) | Mass wt.% per ton of MSW (%) |
|---|---|
| Char ash | 26–45 |
| Flying ash | 1–5 |
| Syngas with acidic components | 1.5–4 |
| Syngas with semi-dry acid | 1–6 |

In solid waste incinerators, only thermal energy is produced, which is utilized in steam production [49,50]. On the other hand, landfilling technologies include open dumps, conventional landfills, and landfill sites equipped with energy recovery, bioreactor systems, semi-anaerobic reactors, as well as flushing bioreactor landfills [1–10]. Landfilling relies on methane gas capture through the anaerobic digestion of biowaste for electricity production. Most landfills and incinerators maintain the following process stages [49]:

- Processing of municipal solid waste.
- The collection system of unprocessed deposits, such as metals, wood, polymers, and organic materials, using magnetic separators.
- High-temperature incinerators with excess oxygen supply.
- Ash removal system and separators.

- Steam and electricity generation.
- Ash disposal systems.

In landfill sites, leachate and biogas are extracted by methane generation in underground anaerobic digestors. Emission control and energy recovery from hydrocarbon gaseous release are also included. As shown below in Table 6, $NO_2$, dioxins, and carbon dioxide emissions are much higher for incineration in comparison with landfilling. Moreover, incineration emits 66% more $CO_2$ than landfilling, making it environmentally unviable.

**Table 6.** Environmental emissions from incineration and landfilling [49].

| Gaseous Component | Incineration (g/T) | Landfilling (g/T) |
|---|---|---|
| Nitrogen oxide | 1600 | 680 |
| Particulates | 39 | 5.3 |
| $SO_2$ | 42 | 53 |
| HCl | 58 | 3 |
| HF | 8 | 3 |
| VOCs | 8 | 6.4 |
| Cadmium | 0.005 | 0.071 |
| Nickel | 0.05 | 0.0095 |
| Arsenic | 0.005 | 0.0012 |
| Mercury | 0.05 | 0.0012 |
| Dioxins and furans | $4 \times 10^{-7}$ | $1.4 \times 10^{-7}$ |
| Carbon dioxide | 1,000,000 | 300,000 |

## 2. Environmental Assessment of Greenhouse Gas Emissions from Thermochemical Reactions

Several approaches exist for measuring greenhouse gaseous emissions, mainly CO, $CO_2$, $N_2O$, and $CH_4$, from thermochemical reactors. Important factors to determine the quality of the flux measurements from thermochemical reactors are the collected gaseous samples for these reactors [21,22,25]. Thermochemical reactions, such as pyrolysis and gasification, burn biomass or solid waste, with insufficient oxygen supply under stoichiometric conditions to produce combustible gaseous products, referred to as syngas. These thermochemical processes are recommended due to the reduced release of toxins, as well CO and methane, compared to those released from the combustion process. In gasifiers, the air-to-fuel ratio varies between 5:1 and 8:1, while the required ratio for combustion is 3:1 [4,21,22,52] There are several methods for controlling and reducing greenhouse gaseous emissions from thermochemical reactions, including increasing energy efficiency, the switching of fuel, heat integration, and the use of more efficient methods, such as heat exchanger networks and the catalytic conversion of $NO_x$ and CO emissions [4,52,53].

The controlled variables are variables that remain constant throughout the reaction, ensuring accurate temperature profiles. Controlled variables are kept constant, so they do not influence the reaction outcomes. Controlled variables could be the agitation rate, feedstock rate, nitrogen supply rate, and reflux ratio [54]. The manipulated variables are variables that are controlled, and this change is based on feedback signals, such as thermal plasma or inductive heater current, product withdrawal flow rate, and cooling water flowrate [46,55–57].

The collected solid waste may contain several components such as organic and decomposable materials that might require separation before the combustion process. Organic and decomposable materials are recommended to be sent to pyrolyzers and gasifiers. Unprocessed products from rectors, such as tar, ash, and char, are sent to landfills, as shown below in Figure 2 [49]:

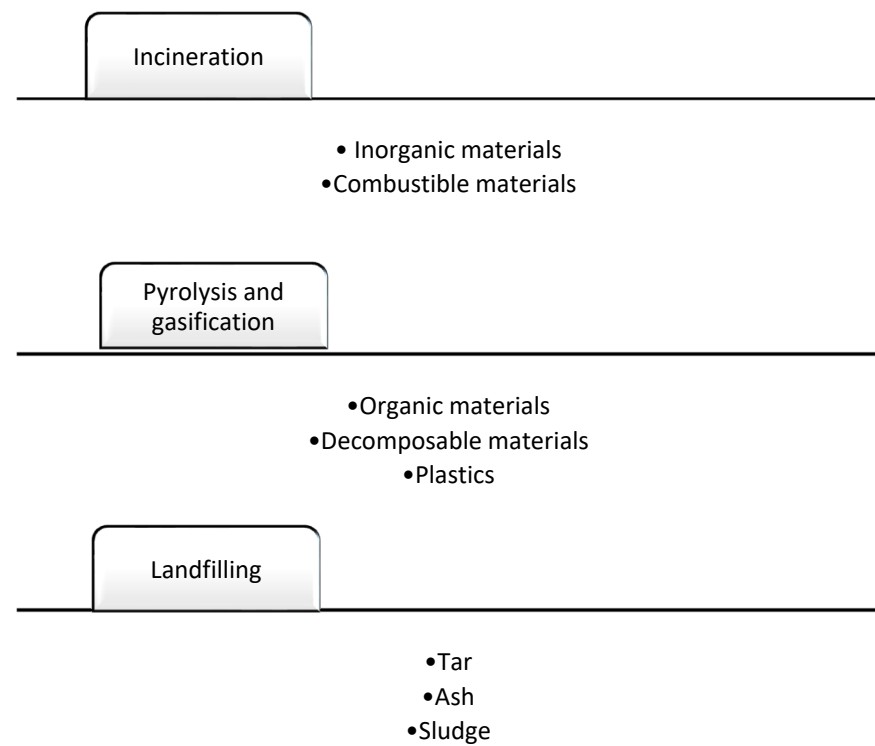

**Figure 2.** Recommended MSW treatment practices for different feedstocks [49].

As seen in Figure 2, organic materials are suitable for pyrolyzers or gasifiers due to their easy conversion and thermal cracking. Incinerators require the separation of incombustible to avoid heat loss and the formation of agglomerates. Landfilling as the last recycling strategy is recommended for slag, unprocessed waste, and other materials that cannot be processed in chemical recycling processes [58]. According to EU regulations, a reduction of 65% is required in landfilling facilities [55,56,59]. In a standard landfilling chemical plant, the following process stages are required [60]:

- Municipal solid waste processing.
- A gas separation and processing unit.
- An environmental control and monitoring unit.
- A gas and steam combustion unit.
- A steam generation unit.
- A waste to energy process system.
- A heat integration unit.

The thermochemical processes have limiting factors, such as high thermal energy consumption, high environmental impact, and low thermal efficiency, as well as the release of greenhouse gas emissions [49]. Optimal boiler conditions and turbine efficiency are required for high-energy generation. Incineration is the full combustion of heterogenous combustible matter, in excess oxygen, of organic and inorganic matter, including minerals with the highest allowable water vapor content of 35 wt.% and an optimal moisture content of around 15 wt.%. Below are the main process stages in incinerators:

Drying and degassing stage: This stage prepares municipal solid waste with optimal moisture content and surface area to ensure optimum heat transfer and low water content, which helps improve energy efficiency. This stage reduces also the PSD (particle size diameter) of MSW feedstock, which aids in heat transfer and helps avoid agglomeration and slug formation.

Incinerator stage: A thermal cracking process of MSW, in excess oxygen, releasing thermal energy at 800 °C to 1000 °C [42,51]. The solid waste volume is massively reduced, and the volume of undesired products, including tar and char content, is minimized.

Incinerators usually contain several heat zones and two air supply sources to ensure the complete combustion of combustible solid waste materials. The released combustible gas contains dioxins, furans, nitrogen oxides, carbon monoxide, and oxygen in controllable levels, depending on the municipal solid waste mass composition. Excess oxygen is supplied to ensure a complete combustion process, based on stoichiometric calculations. Incinerators adapt different grate designs and heat transfer surface areas, depending on the MSW feedstock and the heat exchanging network (HEN).

Flue gas scrubbers: This stage focuses on the removal of slug, as well as unprocessed waste and heavy metal contaminants, during the incineration process to ensure that the incinerator complies with environmental standards before releasing flue gas into the atmosphere.

Boiler and steam generation stage: Combustible gases, including syngas and light hydrocarbon gases, are burnt using a gas and steam turbine to optimize steam and electricity generation. The steam is generated in a heat recovery network using flue gas and a steam generation cycle.

The incineration quality is determined by the degree of complete combustion, which could be measured by the mass percentage of CO, $CO_2$, and $NO_x$, since complete combustion requires negligible carbon monoxide levels below 5 ppm [49]. The residence time of solid waste incinerators is from several minutes to one hour, based on the mass composition of the solid waste feedstock and the process temperatures used [36]. Incinerators ensure minimum combustion temperatures and minimum residence time to ensure full combustion in the primary and secondary air zones, including excess oxygen supply [49].

Boilers integrated in incinerators could have different designs including vertical or horizontal setups with different oxygen levels [36]. A typical incinerator steam generation network is divided into a superheater, an economizer, and an evaporator. Pyrolysis requires inert conditions provided by a nitrogen or argon gas supply at elevated temperatures, as shown in Table 1. Thus, the following equation illustrates the thermal cracking and energy generation process of incinerators [36]:

$$C_nH_m + heat \rightarrow bCO_2 + cCO + fossil\ fuel\ oil + tar + dH_2O$$

## 3. Recommended Monitoring and Process Control Techniques for Thermochemical Reactions

For the provision of continuous data from measurement analyzers and sensors, intelligent control systems and optimization are required in the field of thermochemical processes to reduce operating costs and increase overall process efficiency [61–64]. Numerical process modeling, based on theoretical principles, and process validation with relevant parameters provides continuous monitoring of greenhouse gaseous emissions, which offers continuous feedback to process control systems and adjusts thermal control equipment, thus improving overall process efficiency and reducing the production of tar and char [65–69]. The adjustment of reactor systems requires operational experience and continuous measurement of process parameters [70–72].

The combination of the data-driven approach with the physical principles of thermochemical processes could be the solution for better understanding and control of thermochemical process systems. An expected flow diagram of an AI-based hybrid control system that includes the measurement of process data, data processing, and decision making, as well as machine learning (ML) algorithms with metrics for the automation and evaluation of the process parameters and the measurement of greenhouse gaseous emissions during thermochemical reactions, is shown below in Figure 3:

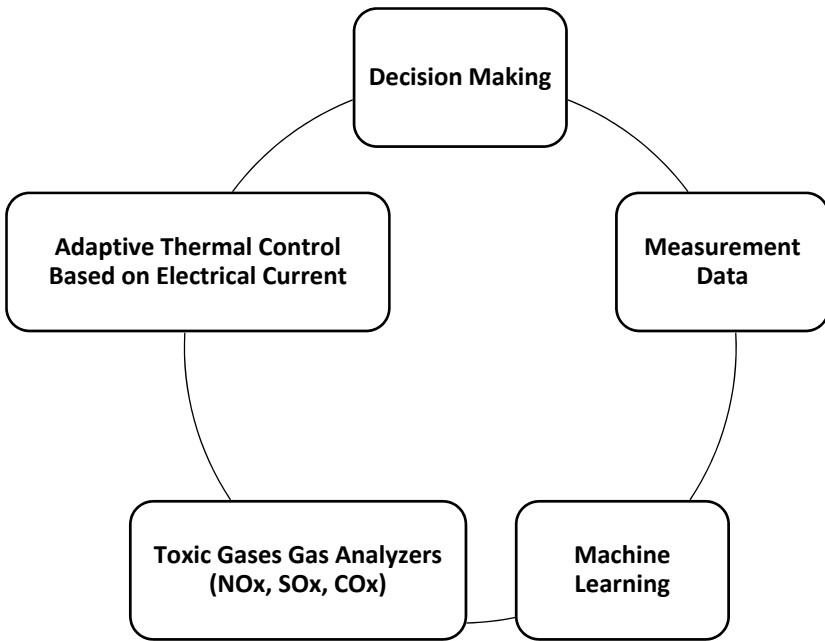

**Figure 3.** Procedure for the continuous monitoring of toxic gaseous emissions during thermochemical reactions for thermal control [73–75].

The monitoring and process control measurement algorithms enable robust and accurate predictions for optimal process control and gas monitoring techniques, enabling online and robust predictions for the control of the thermal source during thermochemical reactions for optimal process control, as shown below in Figure 4 [76–78]:

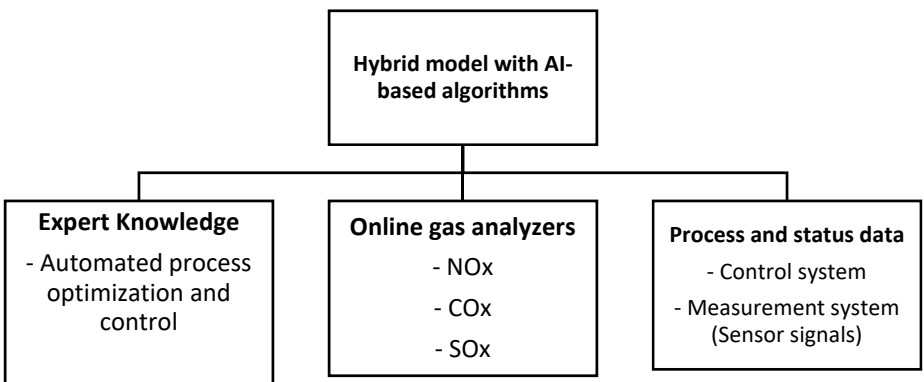

**Figure 4.** Hybrid model for the online measurement and process control improvement for the reduction of greenhouse gaseous emissions in thermochemical reactions.

For the measurement of hydrogen during thermochemical processes, hydrogen gas analyzers, such as the Yokogawa gas density hydrogen analyzer, measure the purity and mass concentration of hydrogen in real time, as well as moisture content, without the need for water removal [48,64,79,80]. Common gas analyzers can also monitor $O_2$ concentration to determine whether the reactor is gas-tight and suitable for pyrolysis reactions. Additionally, common gas analyzers can measure CO, $H_2S$, methane, ammonia, and other gaseous emissions in real time during the thermochemical process.

A common strategy in the control of thermochemical processes includes lowering the air/oxygen-to-fuel ratio of the process, which ensures gasification and prevents solid waste combustion, thus preventing the generation of greenhouse gaseous emissions [1–20]. The critical parameter in the process is the air-to-fuel mass ratio, which ensures the gasification reaction occurs and prevents the combustion process. Improving the process control over

the gasifiers and the process optimization tools also prevents the generation of greenhouse gaseous emissions during the thermochemical process [81–85].

Process monitoring tools such as multivariable process analyzers used for fault detection and online diagnosis of greenhouse gaseous emissions also ensure process safety, process reliability, high production rates, and high product yields, with fewer undesired products. Multivariable statistical process controls and monitoring analyzers are used for continuous fault diagnosis in chemical reactors. The monitoring and process control of chemical reactors aim to improve process selectivity, minimize raw material consumption, improve the quality of the final products, and reduce greenhouse gaseous emissions and undesired products [10–30]. Process control and monitoring tools are also used in the optimization of separation techniques for determining the composition of mixtures and process constraints that affect the product yield.

Through effective process control, the reduction of greenhouse gaseous emissions includes control over energy efficiency, fuel switching, and the efficient use of recycled materials. The implementation of effective process monitoring, and control strategies can control carbon emissions at various process levels to formulate necessary strategies to manage the emissions of greenhouse gases and undesired products. The process control loops will control pressure and temperature and maintain the concentration and safety of process parameters, preventing runaway reactions during thermochemical processes. Controlling process parameters, such as heat release, operating temperatures, and chemical concentrations, reduces greenhouse gaseous emissions and eliminates the production of tar and char.

The reaction temperature is a controlled variable that controls the reaction rate, side reactions, and the distribution of the final products. The reaction temperature is detected using sensors, and the flow of cooling fluid is controlled using monitoring and flow controllers that control the process parameters. For the control of batch reactors, the superior performance of advanced control techniques is recommended over PID control systems. Moreover, model predictive control (MPC) is the recommended control strategy that predicts future process outputs and the optimization of process parameters through controlling input and output process constraints. The tuning method includes controlling the weighting parameters, prediction, and control strategies, and tuning the process parameters to handle constraints and achieve plant stabilization.

## 4. Discussion

This work presents recommended monitoring and gas analysis tools to continuously monitor greenhouse gaseous emissions based on developed mathematical models that analyze the thermochemical reaction behavior in real time and thus provide feedback to the thermal control system, reducing greenhouse gaseous emissions by providing accurate thermal control. Online monitoring equipment can improve operational factors such as thermal efficiency through feedback control that limits the thermal source and thus saves thermal energy and improves energy management of thermochemical reactors. Greater thermal control, with the help of online monitoring equipment, helps improve the reaction kinetics of thermochemical reactions such as pyrolysis and gasification, which are highly dependent on thermal control. Recommended control strategies for thermochemical reactors include the implementation of advanced control techniques such as PID controllers and model predictive control (MPC), which improve temperature profiles and eliminate undesired products, as well as greenhouse gaseous emissions.

In terms of real-time gas analysis, for hydrogen gas, Yokogawa hydrogen gas analyzers are recommended to monitor gasification reactions. The oxygen level is a critical parameter in pyrolysis and gasification, since it determines the composition of the final products. In pyrolysis, hydrogen sulfide and carbon monoxide gas analyzers are required during pyrolysis as a safety measurement, and they provide personnel with an adequate warning of hazardous exposure, as a safety precaution.

The procedure for adaptive thermal control to eliminate emissions during thermochemical reactions starts with decision making, following by measurement data and machine learning techniques to provide feedback control to thermal control systems. Toxic gas analyzers, sensitive to $NO_x$, $SO_x$ and $Co_x$, are used to provide online feedback control signals to adaptive thermal control systems to improve thermal performance and eliminate the production of tar and greenhouse gases. Online gas analyzers are also useful tools for maintaining the air-to-fuel ratio, as well as the oxygen mass ratio, during thermochemical reactions. In gasifiers, the air-to-fuel ratio varies between 5:1 and 8:1, while the required ratio for combustion is achieved using feedback control signals and oxygen gas analyzers during gasification reactions.

For the measurement of hydrogen during thermochemical processes, hydrogen gas analyzers, such as the Yokogawa gas density hydrogen analyzer, measure the purity and mass concentration of hydrogen, as well as moisture content, in real time, without the need for water removal [48,64,79,80]. Common gas analyzers can also monitor $O_2$ concentration to determine whether the reactor is gas-tight and suitable for pyrolysis reactions. Additionally, common gas analyzers can measure CO, $H_2S$, methane, ammonia, and other gaseous emissions in real time during the thermochemical process.

A common strategy in the process control of thermochemical processes includes lowering the air/oxygen-to-fuel ratio of the process, ensuring gasification and averting solid waste combustion, thus preventing the generation of greenhouse gaseous emissions [1–20]. The critical parameter in the process is the air-to-fuel mass ratio, which ensures that the gasification reaction occurs and prevents the combustion process. Improving process control over the gasifiers and process optimization tools also prevents the generation of greenhouse gaseous emissions during the thermochemical process [81–85].

Process monitoring tools such as multivariable process analyzers used for fault detection and online diagnosis of greenhouse gaseous emissions also ensure process safety, process reliability, high production rates, and high product yields, with fewer undesired products. Multivariable statistical process control and monitoring analyzers are used for continuous fault diagnosis in chemical reactors. The monitoring and process control of chemical reactors aim to improve process selectivity, minimize raw material consumption, improve the quality of the final products, and reduce greenhouse gaseous emissions and undesired products [10–30]. Process control and monitoring tools are also used in the optimization of separation techniques for determining the composition of mixtures and process constraints that affect the product yield.

Thus, the proposed control strategy aims to eliminate greenhouse gas emissions by providing real-time control in pyrolysis and gasification reactions [68,86]. The reaction kinetics of the thermochemical process mainly depends on the process temperature. The kinetic reaction and gaseous emissions are incorporated into numerical models to describe the thermochemical conversion behavior and provide feedback control of the thermochemical process to reduce greenhouse gaseous emissions and increase the quality and conversion yield of hydrocarbon liquid and gaseous products.

There are more than 100 process parameters in the thermochemical process, yet only a few can be controlled by operators. Process analyzers, such as gas analyzers, improve process reliability and provide optimal conditions for the thermochemical process. The main challenge for controlling thermochemical reactors relates to controlling energy conversion efficiency, energy losses during the thermochemical process, and thermo-mechanical stress. The optimization procedure is carried out by optimizing the receiver shape and dimensions, the mode of the reactant feed, and the particle morphology during the thermochemical process.

The four operational parameters, including operating temperature, air-to-nitrogen ratio, steam-to-feedstock ratio, and inlet gas velocity, are required to optimize and analyze the condition parameters for various thermochemical processes. Moreover, intelligent evaluation and optimization tools are important parameters in thermochemical reactors to reduce cost and increase process efficiency, as well as to eliminate greenhouse gaseous

emissions. The reduction of greenhouse gaseous emissions could be improved by increasing the process efficiency, fuel switching, combining heat and power, and the heat integration of thermochemical processes. Process efficiency could also be improved by optimizing pyrolysis temperature and changing feedstock, which helps in reducing $CH_4$ and $N_2O$ emissions. Introducing advanced thermal control systems, such as thermal plasma and conductive heaters, also contributes to the reduction of greenhouse gaseous emissions by achieving accurate temperature profiles and preventing the generation of tar and char.

The capture, monitoring, and process measurement of thermochemical processes with process data and the modeling of process parameters of thermochemical processes aid in decision making and achieving higher process control, improving the efficiency of energy generation. Thermochemical processes coupled with power-to-heat techniques, have shown better performance in comparison to latent heat storage technologies in terms of storage time dynamics and energy density. Improving the quality of gaseous products is achieved by increasing the efficiency of the pyrolysis reactor and providing continuous measurement and control over the thermal source to reduce greenhouse gaseous emissions.

Process control strategies are employed to control common process variables such as reactant flow, liquid pressure, and operating temperature of endothermic reactions. The four input process parameters are: input flow rate, utility flow rate, operating temperature, and pressure. Process parameters, such as feed composition and impurity level, also have a direct impact on the product yield. Other parameters that could be controlled are product and effluent flow rates and their operating pressures, temperature, and reactant chemical composition. Controllable process parameters, such as the operating temperature of the distillation towers, reactor temperature, and pressure, also have a direct impact on the process flow. The main components of the control system are the measuring element, the controller, and the final control element.

With an online process control that detects greenhouse gaseous emissions and process parameters of thermochemical processes, the feedback control system adjusts the thermal source and provides accurate temperature profiles that eliminate undesired products such as tar and char. For example, in incineration and gasification processes, the air-to-fuel ratio, as well as the stoichiometric parameters, have a direct impact on the product yield, as well as the chemical composition of the final products. Online monitoring and control techniques could provide online control over thermal sources such as thermal plasma, and inductive or electric heaters that control operating temperatures and reduce or eliminate tar and char generation.

Online monitoring techniques are used to monitor the chemical process in real time, with analytical instruments that enable real-time monitoring of reactions, as well as provide immediate data regarding critical process control parameters [1–20]. The chemical reactors could be monitored with sensors that can accurately monitor process concentrations, phase separations, and chemical changes in process liquids. For automatic control systems, the measured parameters are transmitted to the process control system using analog or digital outputs via a 4–20 mA signal to control process constraints and maintain concentrations during the batch process. The feedback control system is responsible for the detection of unforeseeable disturbances and for controlling the manipulated variables to maintain the process limits. The control system consists of proportional action, whereas the controller signal is proportional to the process deviation from the setpoint. The online monitoring of process parameters, including on-stream measurements of process parameters that measure chemical composition in real time, is controlled via derivative and integral actions.

The environmental assessment of thermochemical reactions includes the measurement of air pollutants such as nitrogen oxides, sulfur dioxides, carbon and nitrogen monoxide, particulate matter, heavy metal concentrations, and carbon dioxide during thermochemical processes. Online monitoring tools that measure dioxins and furans can also determine the potential environmental consequences in thermochemical reactors. The online monitoring of thermochemical reactors includes online analytical instruments to monitor the mass ratio of hydrocarbon final products and carbon monoxide and carbon dioxide to evaluate

the performance of the thermochemical process and measure the number of reactants and products in real-time to reveal the state of the chemical reactor, providing online measurements of reactor constraints, including operating temperature, operating pressure, reactor level, fluid density, and liquid viscosity.

Online monitoring techniques contribute to the improvement of product quality and consistency, as well as increase process efficiency and ensure safe operations by providing online monitoring parameters during the process. Optimal control of thermochemical reactors is considered a challenging task, since the manipulation of non-linear, asymmetric, and process uncertainties are needed to control process parameters. The optimization of process parameters ensures offset-free control performance, and process parameters satisfy the process constraints that are calculated based on multiple input constraints.

## 5. Conclusions

In conclusion, online monitoring technologies integrated with PID controllers play a vital role in achieving accurate temperature control, increasing product yield and reducing or eliminating undesired products, such as tar and char. The online gas analyzers provide feedback control to the thermal control system to increase or reduce the operating temperature during the thermochemical process. Online gas analyzers are also able to maintain the oxygen: fuel ratio in gasification, which plays a vital role and ensures oxygen levels are within the acceptable limits.

This approach ensures the reduction or elimination of greenhouse gaseous emissions and undesired productions and provides higher thermal control during the process, as well as online monitoring and measurement of the performance of the thermochemical process. Online monitoring techniques can provide continuous data for online gas analyzers and sensors, increasing both process control and thermal efficiency. Thermal plasma inductive heaters are controlled based on feedback control signals from online gas analyzers that provide feedback control and reduce or eliminate tar content in pyrolysis and gasification reactions. The integration of pyrolyzers and gasifiers into combined heat and power methods has also shown improvement in energy efficiency, achieving complete combustion reactions, and thus reducing greenhouse gaseous emissions. Moreover, the usage of pure feedstock, including biomass such as wood or sawdust, has achieved the elimination of greenhouse gaseous emissions and char and increased the product yield of hydrogen and syngas. New intelligent control methods, such as adaptive thermal control and decision making based on feedback control signals, are used to reduce or eliminate greenhouse gaseous emissions.

**Author Contributions:** Conceptualization, M.A.; methodology, M.A.; formal analysis, M.A.; investigation, M.A.; resources, M.A.; data curation, M.A.; writing—original draft preparation, M.A. and I.M.R.F.; writing—review and editing, M.A. and I.M.R.F.; supervision, I.M.R.F.; project administration, M.A. and I.M.R.F.; funding acquisition, I.M.R.F. All authors have read and agreed to the published version of the manuscript.

**Funding:** This research was funded by the University of Technology Sydney through Strategic Research Support funding with grant number 2200034.

**Institutional Review Board Statement:** Not applicable.

**Informed Consent Statement:** Not applicable.

**Data Availability Statement:** No new data was created or analyzed in this study. Data sharing is not applicable to this article.

**Conflicts of Interest:** The authors declare no conflict of interest.

## Abbreviations

| | |
|---|---|
| CHP | combined heat and power cycle |
| GWP | global warming potential |
| LHV | lower heating value MJ/nm$^3$ |
| HTP | human toxicity potential |
| MSW | municipal solid waste |
| MSWM | municipal solid waste management |
| PSD | particle size diameter |
| RDF | refused derived fuel (i.e., treated MSW feedstock) |
| VOC | volatile organic compound |

## Nomenclature

| | |
|---|---|
| E$_{syngas}$ | energy value (KJ/Kg) |
| K | Kelvin |
| Mg/Rm$^3$ | milligram per dry cubic meter of flue gas |

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
