# Peer review of "Environmental Analysis, Monitoring, and Process Control Strategy for Reduction of Greenhouse Gaseous Emissions in Thermochemical Reactions"

_atmosphere, doi:10.3390/atmos14040655_

Round 1

Reviewer 1 Report

1. The introduction lacks logic. It is suggested to make changes to make readers better understand the author's research content.

2. Line 58: “reactions s as” to “reactions as”.

3. Line 94: “(M.Al-Maaded, 2012).” what is it? The citation seems to be chaotic.

4. Line 369: “main challenge to controlling” to “main challenge of controlling”. Please check the spelling of the paper carefully and correct similar mistakes.

5. Please complete the conclusion, which seems to be inconsistent with the research content stated in the introduction at the beginning.

Author Response

Thank you for your comments.

Q1. The introduction lacks logic. It is suggested to make changes to make readers better understand the author's research content.

Answer: The introduction is revised to ensure coherence and logic

Q2. Line 58: “reactions s as” to “reactions as”.

Answer: Line 58 corrected

Q3. Line 94: “(M.Al-Maaded, 2012).” what is it? The citation seems to be chaotic.

Answer: Line 94 corrected. Other references are included 

Q4. Line 369: “main challenge to controlling” to “main challenge of controlling”. Please check the spelling of the paper carefully and correct similar mistakes.

Answer: corrected

Q5. Please complete the conclusion, which seems to be inconsistent with the research content stated in the introduction at the beginning.

Answer: Conclusion corrected. The manuscript is revised for coherence.

Reviewer 2 Report

The subject matter is important. However, in its current form it cannot be published. Then:

1. The paper is proposed as a review, however, of the 51 cited references, only 1/3 are from the 2020s, a quick search on Science Direct will reveal much more than this.

2. Discussion is not supported by the cited references.

3. The conclusion is too obvious, and also not supported by the discussion.

Author Response

Thank you for your comments.

The subject matter is important. However, in its current form it cannot be published. Then:

1. The paper is proposed as a review, however, of the 51 cited references, only 1/3 are from the 2020s, a quick search on Science Direct will reveal much more than this.

Answer: More references are added from 2020

2. Discussion is not supported by the cited references.

Answer: Discussion is revised

3. The conclusion is too obvious, and also not supported by the discussion.

Answer: Conclusion and overall manuscript is revised

Reviewer 3 Report

The authors set the stage to perform a review study of monitoring technologies and process control tools for various greenhouse gaseous emissions (CO2, NOx, NH3, N2O, CO) of different thermochemical reactions. An environmental assessment study, including inventory analysis, impact and sensitivity analysis for each technology in terms of greenhouse gaseous emissions, heat generation, electricity and methane is conducted. This work also provides recommended analytical equipment and monitoring tools for different thermochemical reactions to optimize process parameters and reduce greenhouse gaseous emissions. This research study is interesting but there are some issues that have to be addressed.   1) In introduction, the authors are referred to thermochemical processes for GHG emissions, heat & electricity generation, but they don't name and describe them. A paragraph (perhaps and a diagram) should be added to describe shortly these technologies.    2) Table 1: This Table should be reconstructed. Apart from products, percentage of GHG emissions, hydrogen and/or syngas production, thermal and electrical efficiency values shall be included. Combustion process has to be added in order to be a fair comparison with the other 2 processes.   3) Section 3 (Monitoring and process control techniques): It is interesting to mention the process control systems/techniques (eg. PI, PID) applied for each thermochemical technology according to literature (in a Table). Controlling variables and/or manipulated variables should also be included.  

Author Response

The authors set the stage to perform a review study of monitoring technologies and process control tools for various greenhouse gaseous emissions (CO2, NOx, NH3, N2O, CO) of different thermochemical reactions.   An environmental assessment study, including inventory analysis, impact and sensitivity analysis for each technology in terms of greenhouse gaseous emissions, heat generation, electricity and methane is conducted.   This work also provides recommended analytical equipment and monitoring tools for different thermochemical reactions to optimize process parameters and reduce greenhouse gaseous emissions.   This research study is interesting but there are some issues that have to be addressed.

Thank you for your comments.

1) In introduction, the authors are referred to thermochemical processes for GHG emissions, heat & electricity generation, but they don't name and describe them.   A paragraph (perhaps and a diagram) should be added to describe shortly these technologies.

Answer: A figure has been added to describe the thermochemical reactions

2) Table 1: This Table should be reconstructed.   Apart from products, percentage of GHG emissions, hydrogen and/or syngas production, thermal and electrical efficiency values shall be included.   Combustion process has to be added in order to be a fair comparison with the other 2 processes.

Answer: Thermal and electrical efficiencies are added in Table 1

3) Section 3 (Monitoring and process control techniques): It is interesting to mention the process control systems/techniques (eg.   PI, PID) applied for each thermochemical technology according to literature (in a Table).   Controlling variables and/or manipulated variables should also be included.

Answer: I have added the expected controlling and manipulated variables for thermochemical reactions (209-214)

Round 2

Reviewer 1 Report

The authors assessed the different thermochemical processes over 15 different environmental indicators and techno-economic key performance indicators (KPI), and illustrated the recommended monitoring technologies for detection of various greenhouse gaseous emissions for solid waste thermochemical reactions including incineration, pyrolysis, and gasification. After revision, the quality of the manuscript has been greatly improved. The following comments are intended to further improve the quality of the manuscript. Details can be viewed in the attachment

Author Response

  • Line 39 corrected
  • Line 41 corrected
  • Line 47 and line 48 corrected
  • Line 55-Line 56 corrected
  • Table 4 included in manuscript

Reviewer 2 Report

The authors improve the relevance and contemporaneity of the references. However, the discussion still needs to be based on the cited references. Thus, as the conclusion must come as a result of the synthesis of the discussions.

Author Response

I have revised the discussion and conclusion as requested. The discussion revised to be based on cited references. The conclusion is revised to be based on the discussion.